



# Parallelizing a serial code: open–source module, EZ Parallel 1.0, and geophysics examples

Jason L. Turner[1] and Samuel N. Stechmann[1,2]

[1]Department of Mathematics, University of Wisconsin–Madison, Madison, WI, USA
[2]Department of Atmospheric and Oceanic Sciences, University of Wisconsin–Madison, Madison, WI, USA

**Correspondence:** Jason L. Turner (jlturner5@wisc.edu)

**Abstract.** Parallel computing can offer substantial speedup of numerical simulations in comparison to serial computing, as parallel computing uses many processors simultaneously rather than a single processor. However, it typically also requires substantial time and effort to convert a serial code into a parallel code. Here, a new module is developed to reduce the time and effort required to parallelize a serial code. The tested version of the module is written in the Fortran programming language, while the framework could also be extended to other languages (C++, Python, Julia, etc.). The Message Passing Interface is used to allow for either shared-memory or distributed-memory computer architectures. The software is designed for solving partial differential equations on a rectangular two-dimensional or three-dimensional domain, using finite difference, finite volume, pseudo-spectral, or other similar numerical methods. Examples are provided for two idealized models of atmospheric and oceanic fluid dynamics: the two-level quasi-geostrophic equations, and the stochastic heat equation as a model for turbulent advection–diffusion of either water vapor and clouds or sea surface height variability. In tests of the parallelized code, the strong scaling efficiency for the finite difference code is seen to be roughly 80% to 90%, which is achieved by adding roughly only 10 new lines to the serial code. Therefore, EZ Parallel provides great benefits with minimal additional effort.

## 1 Introduction

Numerical simulations of climate and other geophysical systems can be very computationally expensive in both memory and time, especially when the domain consists of a large physical region. For many years, a solution to these problems has been parallel computing (e.g., Drake and Foster, 1995), in which several processors are utilized simultaneously to divide the computational load.

However, the process of parallelizing a serial code may require a significant amount of time and effort, especially if one is not already familiar with parallel computing algorithms and platforms, such as CUDA, OpenMP, and the Message Passing Interface (MPI). Furthermore, in some cases it is necessary to make a complete overhaul of the original software to take advantage of multi-processor systems if, for instance, one desires a parallel code that is optimized to the greatest extent possible; even though this is not always the case, it leads to the general impression that parallel codes are complicated. As a result, the process of parallelizing a serial code is typically viewed as a major undertaking.





Development motivation of EZ Parallel (EZP) came from the desire to simplify the process of parallelizing a serial code. The goal of the module is to allow the user to make relatively small changes to their original codes to create a new code that can leverage the benefits of parallelism.

The EZP software should be useful for several different scenarios. For example, simulations of idealized models are commonly used, such as quasi-geostrophic (QG) models for the atmosphere or ocean (e.g., Thompson and Young, 2007; Edwards
et al., 2020a, b), stochastic models of cloud patterns (e.g., Hottovy and Stechmann, 2015; Ahmed and Neelin, 2019; Khouider and Bihlo, 2019), tropical circulation models (e.g., Neelin and Zeng, 2000; Lin and Neelin, 2002; Khouider and Majda, 2005; Lin, 2009), and coupled ocean–atmosphere models for El Niño–Southern Oscillation (ENSO) (e.g., Zebiak and Cane, 1987; Xie and Jin, 2018; Thual et al., 2016, 2019), to name only a few examples. Such idealized models have typically been simulated using serial codes, in large part due to the large time and effort required to parallelize a serial code. The EZP software
can help in these scenarios to bring the benefits of parallelism with a minimal amount of additional effort. These scenarios all share a common setup of a set of partial differential equations (PDEs) on a two-dimensional (2D) or three-dimensional (3D) domain, and the EZP module is designed for such a setup. An example of a setup that would not be immediately suitable for EZP is a discrete element model (DEM) for sea ice floes (e.g., Herman, 2016), since it is a model for individual particles (ice floes), rather than a PDE model; nevertheless, this scenario could possibly also be parallelized with some further modifications
to the EZP module. Also, for some large modeling projects, a more detailed rewriting of a code may be needed beyond the minimalist approach described in the present paper, if fully optimized code is desired; nevertheless, even in this scenario, EZP could provide a starting point for parallelism which could be further refined and optimized. Along these lines, EZP is written using MPI, so it can be used for either shared-memory or distributed memory computer architectures.

A related type of tool is automatic parallelization, which is intended to take a serial code and, without needing the user
to change the code at all, automatically implement parallelization in its execution (Griebl, 2004; Bondhugula et al., 2008; Benabderrahmane et al., 2010; Kraft et al., 2018). Such tools are typically designed to treat a variety of different codes with great generality. In their simplest forms, they detect "for loops" in the code and, if possible, they break up the tasks using multithreading or parallelization. Some commercial tools that use these and other strategies include the Matlab Parallel Computing Toolbox™ and Matlab Parallel Server™ or Matlab Distributed Computing Server™. A disadvantage of Matlab is that it is
often slower than other programming languages (Fortran, C++, etc.) for PDEs with time-stepping. For those other languages, the Intel® Fortran and C++ Compilers also have automatic parallelization capabilities, and so does the freely available GNU Compiler Collection (GCC). The commercial tools are not freely available and therefore may not be an option for many users. Auto parallelization is typically limited to shared-memory computer architectures, which limits the amount of parallelization that can be achieved. Also, for some codes, auto parallelization may not achieve parallelism or may not provide a great amount
of speedup. Despite these limitations, it may be a good choice for some users. However, some users may also have concerns about using a "black box" to automatically parallelize a code. Some users prefer to have some control over and some knowledge of the parallelization process. For such users, EZP can help with parallelization with a minimal amount of user effort, while also engaging the user in the parallelization process, and allowing the user to further optimize the parallelization if desired.





Another alternative option for parallelization is to just put in the effort: (i) to put in the substantial effort to *learn* about MPI and parallelization, and then (ii) to put in the substantial effort to use that knowledge to *implement* the parallelization of the code of interest. EZP can help in this option as well: it can help to rapidly reach the end goal of *implementation*, and then much of the other step (the *learning*) can be pursued later, while the parallel code is already up and running.

The paper is organized as follows. We begin our discussion in Section 2 with the general strategy of parallelizing a serial code, and with a description of the new lines of code that must be added to parallelize it (only 10 new lines for a finite difference code, using the EZP module). Then we describe two example codes—two-level QG model and a tropical precipitation model— and present the performance improvements obtained through parallelization with EZP in Section 3. The examples demonstrate the parallelization of two serial Fortran codes, including one finite difference code and one pseudospectral code, and both versions are available at the GitHub repository

https://github.com/jasonlturner/EZ_PARALLEL_project

We hope that the comments in the programs, the documentation included in the repository, and our discussion here makes the module accessible and easy to modify for more personalized use.

## 2   Parallelizing a serial code

In this section, we describe how to parallelize a serial code using EZP. As a conceptual framework for doing so, we assume that the code, whether serial or parallel, consists of two main stages:

1. Initialization (including reading input files and creating the domain grid), and

2. Time-stepping (including stepping the domain grid forward in time and writing output files).

Algorithm 1 is pseudocode of a serial simulation in this format.

---
**Algorithm 1** Example serial code
---
CALL READ_INPUT
ALLOCATE(grid($N_x$, $N_y$)) {The grid is of size $N_x \times N_y$}
CALL INITIALIZE_GRID {Fill in the initial values of the grid}
**for** $i = 1$ to $numTimesteps$ **do**
    CALL TIMESTEP
    CALL WRITE_OUTPUT
**end for**

---

Algorithm 2 shows pseudocode of the parallelization of Alg. 1, with modifications in red. The goals of the rest of the section are two-fold: (i) explain the small additions necessary to turn the serial code into a parallel one, and (ii) explain what those modifications are doing.





---

**Algorithm 2** Example parallel code

---

CALL READ_INPUT

**CALL CREATE_PARALLEL_SCHEME** {The creation of a *parallel scheme* defines the domain decomposition, and adjusts $N_x$ and $N_y$ to be the size of the local sub-grid, see Subsection 2.1 for further discussion}

ALLOCATE(grid($N_x$, $N_y$))

CALL INITIALIZE_GRID

**for** $i = 1$ to $numTimesteps$ **do**

    CALL TIMESTEP

    **CALL SHARE_SUBGRID_BDRY** {The SHARE_SUBGRID_BDRY subroutine executes the inter-process communication to provide each processor all of the data needed to perform a single time-step, see Subsection 2.3 for further discussion}

    CALL **WRITE_OUTPUT** {The WRITE_OUTPUT subroutine requires minor changes to the name of output files, see Subsection 2.4 for further discussion}

**end for**

---

## 2.1 Domain grid decomposition: CREATE_PARALLEL_SCHEME

In creating the parallel code in Alg. 2, the first new addition is a new line of code to call the CREATE_PARALLEL_SCHEME subroutine of the EZP module. This subroutine will decompose the domain grid, and while this is done automatically for the user, we describe some details of the domain decomposition in what follows, as background information.

In a parallel simulation, the domain grid is divided into several sub-grids, as illustrated in Fig. 1. Each sub-grid is handled by a single processor. A vertical slab decomposition is utilized by EZP. To elaborate on the definition of this grid decomposition, suppose the two-dimensional simulation domain is discretized by a grid with $N_x$ points in the horizontal, $N_y$ points in the vertical, and $n$ be the number of processes used to execute the simulation. We denote variables local to a process with a superscript $(k)$ with $k = 0, \ldots, n-1$.

The domain decomposition in EZP is defined by the creation of what we call a parallel scheme, executed by the CALL CREATE_PARALLEL_SCHEME command in the code. The parallel scheme contains all information needed for the grid decomposition, and various parallel computing functions that we will discuss in the following subsections.

One task of the parallel scheme is to divide the $N_x$ columns of the grid as evenly as possible among the $n$ processors. For example, consider a grid with $N_x = 32$ columns to be divided among $n = 3$ processors. The value of $N_x$ would be adjusted by 95 the CREATE_PARALLEL_SCHEME subroutine to the total number of columns of the local sub-grid. In other words, $N_x$ would be automatically changed to $N_x^{(k)}$, the sum of the number of columns in the sub-grid interior and in the sub-grid boundary. In this example, sub-grids 0 and 1 would have 11 columns in their interior while processor 2 would have 10 columns in its interior (see Fig. 1).

Each sub-grid also has sub-grid boundary points that are also included in each value of $N_x^{(k)}$. The sub-grid boundary points 100 represent information that may be needed from neighboring sub-grids. For example, consider the following discretization of





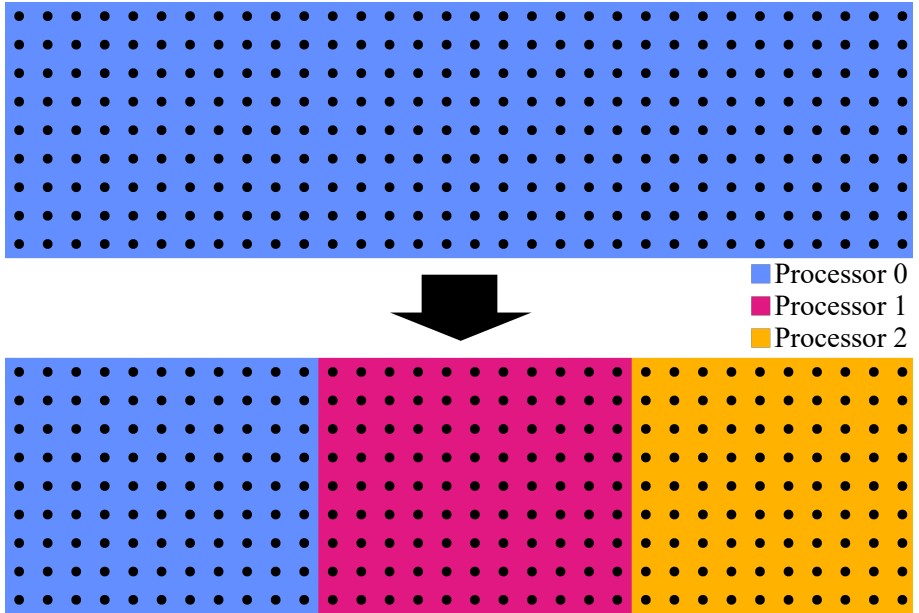

**Figure 1.** The decomposition of a grid (top) into vertical slabs (bottom). Note that we cannot divide the 32 columns of the grid evenly among three processors, so the grid is decomposed as nearly evenly as possible, with the two sub-grids with 11 columns, and one sub-grid with 10 columns.

the 2D Laplacian operator

$$\nabla^2 q = \frac{1}{(\Delta x)^2}\left(q_{i,j-1} - 2\,q_{i,j} + q_{i,j+1}\right) + \frac{1}{(\Delta y)^2}\left(q_{i-1,j} - 2\,q_{i,j} + q_{i+1,j}\right) + \mathcal{O}\left(\Delta x^2\right) + \mathcal{O}\left(\Delta y^2\right). \tag{1}$$

As illustrated in Fig. 2, points on the right-most column of a sub-grid (aside from the right-most sub-grid) will require information from the neighboring sub-grid, and likewise for points on the left-most column of a sub-grid. We will elaborate on
sub-grid boundary communication in Subsection 2.3; for now we discuss only the effect of sub-grid boundary points on the value of $N_x^{(k)}$. As each sub-grid requires one column from each of its neighbors to execute the numerical scheme, sub-grids 0 and 2 would have one column in their boundaries while sub-grid 1 would have two columns in its boundaries. Therefore, the sub-grids would have $N_x^{(0)} = 12$, $N_x^{(1)} = 12$, and $N_x^{(2)} = 11$ columns, respectively, and the domain decomposition has thus been determined.

**2.2   Sub-grid initial condition and time-stepping: `INITIALIZE_GRID` and `TIMESTEP`**

In this subsection, we explain that a user needs to make *no changes* to either the `INITIAL_GRID` or `TIMESTEP` subroutines to turn the serial Alg. 1 into the parallelized Alg. 2.

After creating the parallel scheme, the next step in the code is to define the initial conditions. The `INITIAL_CONDITION` subroutine in the original serial code must be of the form described in Alg. 3 in order to be parallelized with EZP. Specifically,





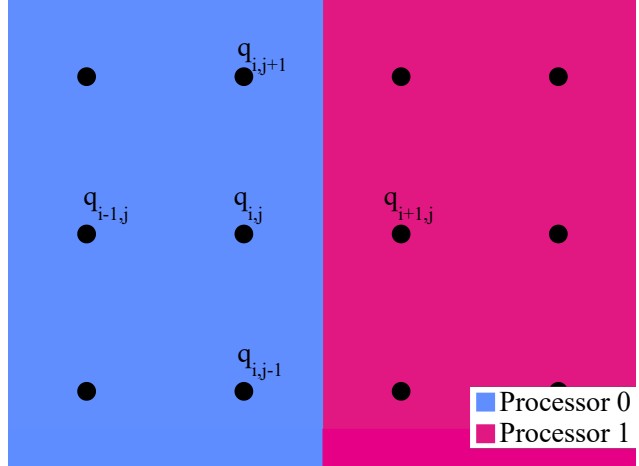

**Figure 2.** The template for the discretization of the Laplacian from Eq. (1) at the boundary of two sub-grids. Due to this discretization, the sub-grid boundary consists of one column from each of its neighbors. Continuing the example from Fig. 1, the sub-grids will then have, including boundary columns, a total of 12, 13, and 11 columns, respectively.

EZP relies on the use of an initial condition function $f(x,y)$ with a reference point $(x_{ref}, y_{ref})$ located in the top-left of the grid. In this form, no changes are required to parallelize the serial code with EZP.

We will now explain how EZP parallelizes the `INITIALIZE_GRID` and `TIMESTEP` subroutines. The `CREATE_PARALLEL_SCHEME` subroutine calculates the position of reference point for the initial sub-grid, and automatically changes the values $x_{ref}$ and $y_{ref}$ to $x_{ref}^{(k)}$ and $y_{ref}^{(k)}$, respectively (see Fig. 3). Hence, all of the grid values in Alg. 3 are

changed to local sub-grid values by a single call to `CREATE_PARALLEL_SCHEME`, and the `INITIALIZE_GRID` subroutine works on the local level. Similarly, all of the grid values in a `TIMESTEP` subroutine are changed to local sub-grid values before it is called, and the `TIMESTEP` subroutine works on the local level as well.

---

**Algorithm 3** `INITIALIZE_GRID` subroutine

---

**Require:** f(x, y) defined
    **for** $i = 1$ to $N_x$, $j = 1$ to $N_y$ **do**
        grid(i,j) ← f($x_{ref} + i * \Delta x, y_{ref} + j * \Delta y$)
    **end for**

---

### 2.3 Sub-grid boundary communication: `SHARE_SUBGRID_BDRY`

The next step in the algorithm is the `SHARE_SUBGRID_BDRY` subroutine. A call to this subroutine is added to serial Alg. 1

to create parallel Alg. 2. While the user needs to only add this one new line of code to call the subroutine, we next describe the tasks of the subroutine as background information.



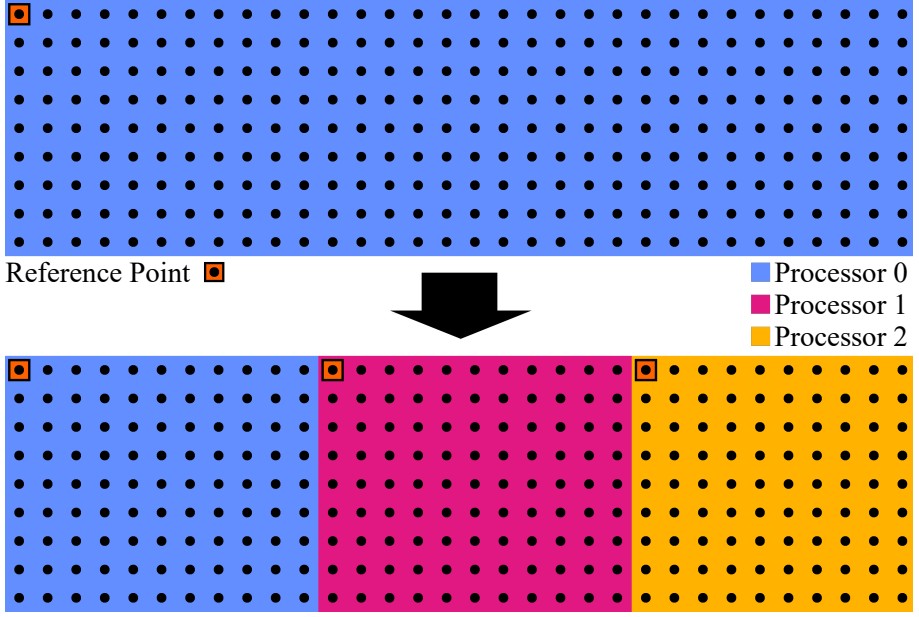

**Figure 3.** The reference point for the grid (top) and the reference points for the sub-grids (bottom) are located in their respective top-left corners. Their location in physical space may be calculated using the widths of the vertical slabs and the spacing between columns of the grid.

In the original serial code (Alg. 1), the `TIMESTEP` subroutine updates the interior of the grid using values from its boundary during each time-step. The `TIMESTEP` subroutine is automatically changed to update the interior of a sub-grid using values from the sub-grid boundary through the creation of a parallel scheme (see Subsection 2.2). Since the boundary of a sub-

grid in the parallel code corresponds to the interior of the grid, it is necessary to update the boundary of the sub-grids at each time-step as well. Since the boundary of one sub-grid corresponds to the interior of a neighboring sub-grid, we may update the sub-grid boundary through inter-process communication. This inter-processor communication is handled by the `SHARE_SUBGRID_BDRY` subroutine, within the EZP module, and it is handled automatically for the user after adding in the one new line of code to call the subroutine.

**2.4    Writing output files: `WRITE_OUTPUT`**

The final stage of the code is the `WRITE_OUTPUT` subroutine of Alg. 1. The user will need to make some minor modifications to their own `WRITE_OUTPUT` subroutine in order to create the parallel version in Alg. 2.

To see the changes that need to be made to the `WRITE_OUTPUT` subroutine, consider a serial code that outputs a single file named '*output.out*'. Without changes, each processor in the parallelized code would attempt to write its individual output

to '*output.out*' simultaneously, leading to a variety of issues (e.g., they all have the same file name, so one processor may write an '*output.out*' file that overwrites the '*output.out*' file created by another processor). To amend this issue, the parallel





scheme contains an identification number that is unique to each processor. This identification number, also called a processor rank (Message Passing Interface Forum, 2015), is stored in the `procID` member of the parallel scheme. The user will need to change the name of the output file to include the processor rank, and then each processor will write its output to a unique file.

## 2.5 Additional changes for psuedo-spectral codes


If pseudo-spectral numerical methods are used, then some additional changes must be made beyond what was described above for finite difference methods, finite volume methods, etc. In particular, we discuss the three tasks of parallel fast-Fourier transforms (FFTs), spectral derivatives, and zero-padding. The EZP module provides subroutines for accomplishing these three tasks in parallel. To parallelize, a user needs to replace the calls to these three types of subroutines (FFT subroutines, spectral

derivative subroutines, and zero-padding subroutines), now using the parallel subroutines provided by the EZP module. EZP will then automatically handle the parallelization. Some additional details are as follows.

FFTs in many serial codes are handled by a single call to a subroutine which performs the intermediate steps. For example, FFTW has an `FFTW_CREATE_PLAN` subroutine to initialize FFTs and a `FFTW_ONE` subroutine to execute FFTs (Frigo and Johnnson, 2018). To parallelize these, EZP includes `CREATE_SCHEME_FFT` for initializing FFTs, and `EXECUTE_SCHEME_FFT`

and `EXECUTE_SCHEME_IFFT` For executing forward and inverse FFTs. A user may parallelize their FFTs by simply replacing their original FFT-related subroutine calls with these. The parallel FFTs of EZP utilize the one-dimensional FFTs of DFFTPACK (Swarztrauber, 1985; Pumphrey, 1985) as well as the global redistribution algorithm described in Dalcin et al. (2019).

Since the entire grid is contained in a single processor in a serial code, it is straightforward to take a spectral deriva-

tive. All spectral derivatives must be changed to parallelize a serial code, as each sub-grid corresponds to a different range of wave numbers. There are two ways to execute a spectral derivative in EZP: (i) `EXECUTE_SCHEME_SPEC_DRV` and `EXECUTE_SCHEME_ISPEC_DRV`, and (ii) `CREATE_SPEC_DRV`. The former must be initialized with the `CREATE_SCHEME_SPEC_DRV` subroutine, and take the desired forward or inverse spectral derivative of the local sub-grid. The latter returns an array containing the coefficients of the desired spectral derivative of the local sub-grid, providing greater

flexibility.

In pseudospectral codes for simulating turbulence, zero-padding is often used to eliminate the aliasing error which occurs in non-linear functions (Orszag, 1971; Donnelly and Rust, 2005). For the same reason that all spectral derivative must be changed to parallelize a serial code, zero-padding must also be handled differently. To accomplish this, EZP provides the `CREATE_SCHEME_ZERO_PAD` subroutine to create a parallel scheme for a zero-padded grid using a parallel scheme for the

original grid, as well as the `EXECUTE_SCHEME_ZERO_PAD` and `EXECUTE_SCHEME_IZERO_PAD` subroutines to zero-pad and inverse zero-pad the grid.





## 3 Examples and parallel performance

We have utilized EZP to parallelize two example codes to demonstrate its abilities and efficacy: a two-level quasi-geostrophic (QG) equations solver (Subsection 3.1) and a stochastic heat equation solver (Subsection 3.2). The former utilizes a psuedo-
spectral method for solving spatial derivatives and a six-stage adaptive additive Runge-Kutta method for time-stepping, which demonstrates the performance gains of using parallel two-dimensional FFTs over serial ones. The latter utilizes a finite difference method for spatial derivatives and a forward-Euler time-stepping scheme, which demonstrates the performance gains of decomposing the grid into sub-grids which requires sub-grid boundary communication.

### 3.1 Two-level quasi-geostrophic equations

The two-level QG equations serve as a simple but fully non-linear fluid model featuring baroclinic instability in a 2D periodic domain:

$$\frac{\partial q_\psi}{\partial t} + J\left(\psi,\, q_\psi\right) + J\left(\tau,\, q_\tau\right) + \beta\frac{\partial \psi}{\partial x} + U\frac{\partial\left(\nabla^2\tau\right)}{\partial x} = -\frac{\kappa}{2}\,\nabla^2\left(\psi-\tau\right) - \nu\left(-1\right)^{-1}\nabla^{2\,s}q_\psi, \tag{2}$$

$$\frac{\partial q_\tau}{\partial t} + J\left(\psi,\, q_\tau\right) + J\left(\tau,\, q_\psi\right) + \beta\frac{\partial \tau}{\partial x} + U\frac{\partial\left(\nabla^2\psi+k_d^2\psi\right)}{\partial x} = \frac{\kappa}{2}\,\nabla^2\left(\psi-\tau\right) - \nu\left(-1\right)^{-1}\nabla^{2\,s}q_\tau. \tag{3}$$

These equations describe the evolution of the barotropic $q_\psi$ and baroclinic $q_\tau$ modes of the potential vorticity (e.g., Qi and
Majda, 2016). These quantities are also determined by the the corresponding barotropic and baroclinic streamfunctions $\psi$, $\tau$, via $q_\psi = \nabla^2\psi$ and $q_\tau = \nabla^2\tau - k_d^2\tau$. Further, $k_d$ is the baroclinic deformation wavenumber, $\beta$ is the Rossby parameter, $J\left(A,\, B\right) = \dfrac{\partial A}{\partial x}\dfrac{\partial B}{\partial y} - \dfrac{\partial A}{\partial y}\dfrac{\partial B}{\partial x}$ is the Jacobian operator, $U$ represents a large-scale vertical shear (equal in strength with opposite directions for each mode), $\nu\nabla^{2\,s}q_i$ represents the hyperviscosity, and $\kappa\nabla^2\left(\psi-\tau\right)$ represents Ekman friction on the lower level of the flow. For more details on the two-level QG equations, see Vallis (2006); Qi and Majda (2016).

The implementation of the numerical method is as follows. The pseudo-spectral method uses Eqs. 2, 3 transformed into spectral space, which changes all spatial derivatives into scalar multiplications. The hyperviscocity, Ekman friction, and mean shear terms are handled in spectral space, and then the non-linear multiplications in the Jacobian terms are handled in physical space (utilizing zero-padding for de-aliasing). We utilize these solutions in each stage of an adaptive additive Runge-Kutta method (AARK) to progress forward in time. The AARK method is made up of ARK4(3)6L[2]SA - ESDIRK for the hyperviscocity
term (which acts on fast time scales) and ARK4(3)6L[2]SA - ERK for the Ekman friction, mean shear, andf Jacobian terms (which act on slow time scales) (Kennedy and Carpenter, 2003). Excluding `USE` statements and MPI initialization and finalization, approximately 85 lines of code were changed between the serial solver and the parallel solver. Most of these changes involve simple replacements, as described in subsection 2.5; for instance, every call to a serial FFT subroutine is replaced by a call to the parallel FFT subroutine provided in the EZP module.

For our own tests, we utilized the high performance computing (HPC) cluster that is serviced by the University of Wisconsin-Madison's Center for High Throughput Computing to investigate the scaling performance of the implementation of the EZP library to parallelize a serial two-level QG equation code. A dedicated partition was used, whose 35 allocatable nodes have a



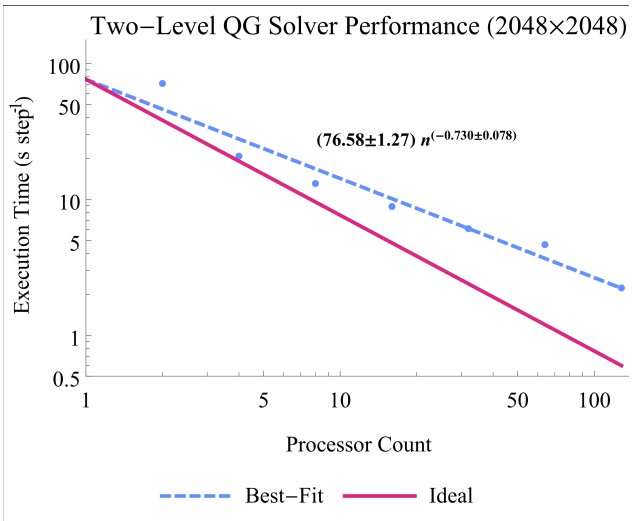

**Figure 4.** Performance of the parallelized QG solver. The average per-step execution time is plotted for the parallelized two-level QG equations solver with $N_x = N_y = 2048$ and $10^3$ time-steps. Reported are the maximum per-step execution times over ten trials. Included is a best-fit power law of the per-step execution time generated using the `LinearModelFit` function of Mathematica® and a line representing the 'ideal' execution time scaling of $cx^{-1}$ for a constant $c$.

single 10 core (20 thread) Intel Xeon E5-2670 v2 CPU at 2.50 GHz, allowing for a total of 700 processes to be run simultaneously.

Simulations were run with a random initial condition on a grid with dimensions $N_x = N_y = 2048$ and $10^3$ time-steps, using a range of two to 128 processors, utilizing eight processors per node for runs of eight or more processors. We chose not to utilize all 20 available processors per node due to the known lack of optimization of `MPI_ALLTOALL` on shared-memory systems (Kumar et al., 2008). Due to the memory-intensive nature of the six-stage AARK method used, there was not enough memory available for a single-processor run with $N_x = N_y = 2048$. The physical parameters in the simulation correspond to

the atmosphere in mid-latitudes, specifically: $\beta = 2.5$, $k_d = 4$, $U = 0.2$, $\kappa = 0.05$, and $\nu = 2.98023 \times 10^{-22}$ (Qi and Majda, 2016). File output was disabled for these runs, and we analyzed the average execution time per time-step for consistency between runs. The results of these runs are included in Fig. 4.

     The observed strong scaling efficiency is $73.0\% \pm 7.8\%$, which implies that doubling the number of processors leads to an approximate 40% reduction in execution time. The difference between the observed scaling and the "ideal" scaling may be

attributed to the non-parallelizable portions of the code (Amdahl, 1967), e.g., inter-processor communication and serial one-dimensional FFTs. Since "ideal" scaling would provide a 50% reduction in execution time (per each doubling of the number of processors), the EZP performance of 40% reduction is somewhat close to ideal, especially in considering that it is achieved without a major overhaul of the serial code structure.





## 3.2 Stochastic heat equation

As an example of a different type of model, a tropical precipitation and water vapor model (Hottovy and Stechmann, 2015) is used, and it takes the form of a modified stochastic heat equation

$$\frac{\partial q}{\partial t} = b_0 \nabla^2 q - \frac{1}{\tau}(q - q^*) + F + D_* \dot{W}, \tag{4}$$

where $q(t, x, y)$ is the integrated column water vapor at horizontal location $(x, y)$, $b_0$ is a spatial interaction constant $\left(\mathrm{h}^{-1}\right)$, $\tau$ is the relaxation time (h), $q^* = 65\,\mathrm{mm}$ is the relaxation target, $F$ is external forcing $\left(\mathrm{mm\,h}^{-1}\right)$, $D_*^2$ is the stochastic forcing

variance $\left(\mathrm{mm}^2\,\mathrm{h}^{-1}\right)$, and $\dot{W}(t, x, y)$ is spatiotemporal white noise. This linear equation is designed to capture idealizations of precipitation, evaporation, and turbulent advection-diffusion of water vapor. Although it is simplistic, the model displays a variety of characteristics that conform to a statistical physics perspective of tropical convection, such as a fractal, scale-free distribution of cloud cluster sizes (Hottovy and Stechmann, 2015; Stechmann and Hottovy, 2016). It may be shown that this equation exhibits infinite variance at the stationary state if $\dot{W}(t, x, y)$ is not regularized. However, the discretization of the

domain accomplishes this. Such types of models are also used in oceanography for modeling sea surface height variability (e.g., Samelson et al., 2016).

To solve this model numerically, we utilized a centered-difference approximation of spatial derivatives with a forward-Euler discrete temporal derivative to reformulate Eq. (4) as

$$q_{i,j}^{k+1} = q_{i,j}^k + \Delta t \left[ b_0 \left( \frac{1}{\Delta x^2}\left(q_{i-1,j}^k - 2q_{i,j}^k + q_{i+1,j}^k\right) + \frac{1}{\Delta y^2}\left(q_{i,j-1}^k - 2q_{i,j}^k + q_{i,j+1}^k\right) \right) - \frac{1}{\tau}\left(q_{i,j}^k - q^*\right) + F + D_* \dot{W}_{i,j}^k \right], \tag{5}$$

for spatial grid points $i, j = 1, \ldots, N$ and time-step $k$. The quantities with subscripts $i, j$ and superscript $k$ are the discrete version of their continuous counterparts in Eq. (4) evaluated at the $(i, j)^{\text{th}}$ column of the atmosphere and time-step $k$, while $\Delta x$, $\Delta y$, and $\Delta t$ are the horizontal, vertical, and temporal grid spacing. Unlike Hottovy and Stechmann (2015), we utilize Dirichlet boundary conditions for simplicity. Excerpts of a generalized version of this code are included in Appendix A.

A dedicated partition of the HPC cluster at the University of Wisconsin-Madison's Center for High Throughput Computing

was also used to investigate the scaling performance of the implementation of the EZP library to parallelize a serial stochastic heat equation code. Simulations were run with a grid size of $N_x = N_y = 8192$ and $10^3$ time-steps, using a range of one to 128 processors, utilizing eight processors per node for runs of eight or more processors. File output was disabled for these runs. The results of these runs are included in Figure 5.

The observed strong scaling efficiency was $86.9\% \pm 1.2\%$, which implies that doubling the number of processors leads to

an approximate $45\%$ reduction in execution time. The difference between the observed scaling and the "ideal" scaling may be attributed to the non-parallelizable portions of the code. With "ideal" scaling, one would obtain a $50\%$ reduction in execution time. Overall, a parallel performance of a $45\%$ reduction is nearly ideal, and it was achieved by changing only roughly 10 lines of code between the serial solver and the parallel solver.

Utilizing the parallelized version of the code, we were able to generate data of a very large system for a long period of time,

specifically $N_x = N_y = 8192$ with uniform grid spacing $\Delta x = \Delta y = 5\,\mathrm{km}$ and $2^{24}$ time-steps with uniform time-steps of size





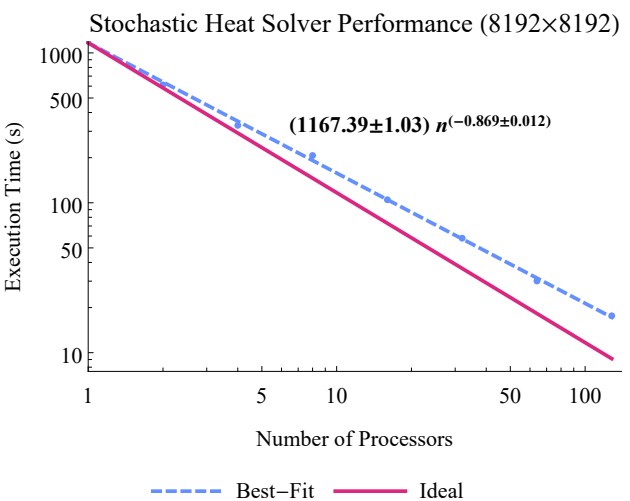

**Figure 5.** Performance of the parallelized finite-difference code. The execution time for the parallelized stochastic heat solver is for $N_x = N_y = 8192$ and $10^3$ time-steps. Reported are the maximum execution times over ten trials. Included is a best-fit power law of the execution time generated using the `LinearModelFit` function of Mathematica® and the line representing the 'ideal' execution time scaling $cx^{-1}$ for a constant $c$.

$\Delta t = 10^{-4}$ h. For comparison, the serial version of the code was used to generate data of a much smaller system for a shorter time, specifically $N_x = N_y = 1024$ and $2^{22}$ time-steps with the same grid spacing and time-step size. Figures 6, 7 show the final state of the simulations, as well as the cloud formations as dictated by the cloud indicator function $\mathcal{H}\left(q_{i,j}\left(t\right) - q^*\right)$ where $\mathcal{H}$ is the Heaviside function.

## 4    Conclusions

In this paper, an open-source module was presented for easing the process of parallelizing a serial code. The module can be used for parallelizing PDE solvers that use a variety of different methods (finite-difference, finite-volume, psuedo-spectral, etc.), and it uses MPI Fortran for applicability to shared-memory or distributed memory computer architectures. The performance gains through its utilization were shown with two use cases, as examples of atmospheric and/or oceanic dynamics models: a two-level QG solver and a stochastic heat equation solver. The parallelization of the former required a change to approximately 85 lines of code, most of which are simple replacements, and parallel code exhibited a strong scaling efficiency of $73.0\% \pm 7.8\%$. The latter required a change to approximately 10 lines of code, and exhibited a strong scaling efficiency of $86.9\% \pm 1.2\%$.

With the development of this module, we have provided some tools that will be of use for parallelizing serial codes. In the future, we hope to implement the following features to increase performance and functionality of EZP: (i) different default domain decompositions, e.g., horizontal slabs, or rectangular chunks not containing an entire column or row of the grid, (ii) native compatibility with domains of different dimensionalities, e.g., three-dimensional grids, $n$-dimensional parallel FFTs, (iii)





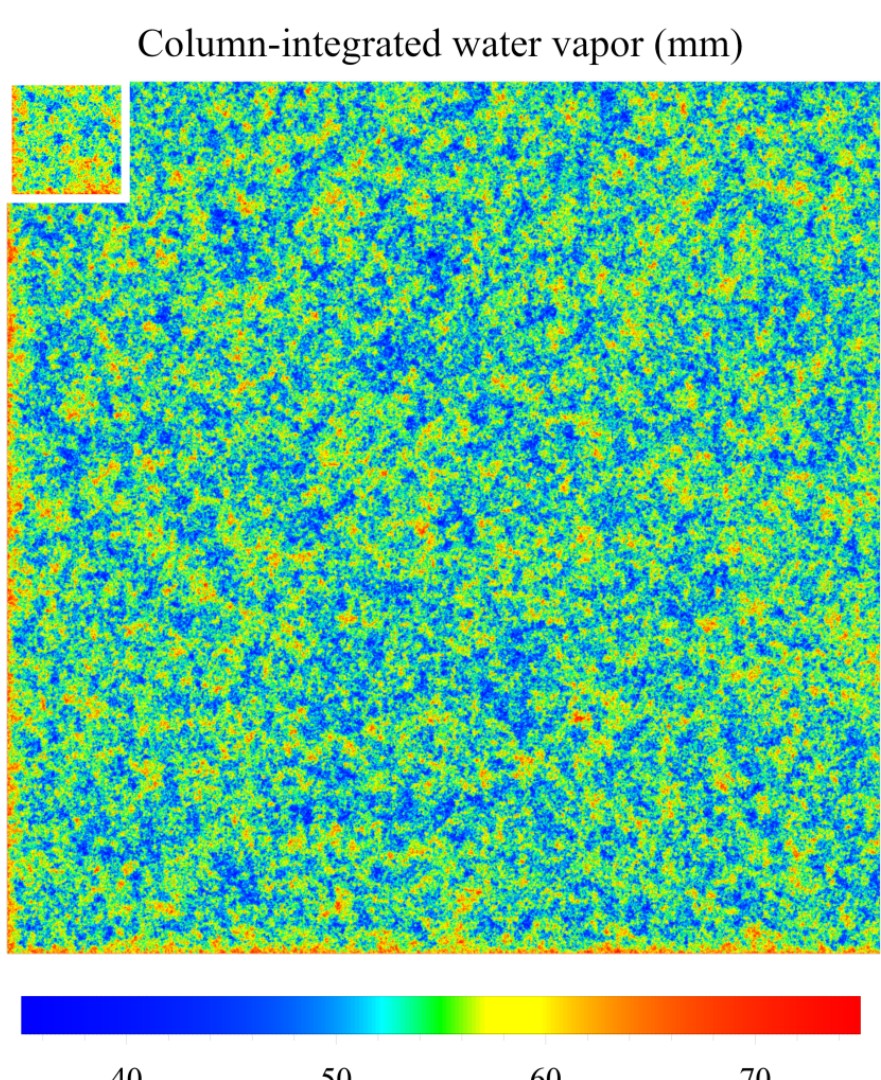

**Figure 6.** The final state of the large and small simulations tropical precipitation model based on the stochastic heat equation, with the smaller grid overlaid in the top-left corner of the larger grid. The larger grid was of size $8192 \times 8192$ grid points while the smaller was of size $1024 \times 1024$, both with a uniform grid spacing $N_x = N_y = 5$ km. The larger simulation was over $2^{24}$ time-steps, while the smaller was over $2^{22}$ time-steps, both with a constant time-step size of $10^{-4}$ h for a total simulation time of approximately 70 and 17.5 days, respectively. Both simulations were initialized at the statistical steady state, as described by Equations (15) and (16) of Hottovy and Stechmann (2015).

expanded FFT options, e.g., utilizing parallel one-dimensional FFTs for transforms along dimensions split between processors, and (iv) compatibility with the C and C++ languages. We have begun the development process for EZP in the C++ programming language, although it has not yet undergone rigorous testing. It would be interesting to also pursue similar software in other

languages, such as Julia and Python.



## Cloud cover



**Figure 7.** The final cloud distribution of the tropical precipitation model based on the stochastic heat equation on a with clouds (white) appearing at grid-points with integrated column water vapor greater than the relaxation target $65$ mm, with the smaller grid overlaid in the top-left corner of the larger grid. The larger grid was of size $8192 \times 8192$ grid points while the smaller was of size $1024 \times 1024$, both with a uniform grid spacing $N_x = N_y = 5$ km. The larger simulation was over $2^{24}$ time-steps, while the smaller was over $2^{22}$ time-steps, both with a constant time-step size of $10^{-4}$ h for a total simulation time of approximately 70 and 17.5 days, respectively.

Further, the EZP module is not limited to 2D geophysical systems and could be used more broadly to expand the scope of various computational projects. Although it was presented here for 2D examples, it can be used for systems of various dimensionalities. For example, one could declare a one-dimensional grid as a 2D grid with a single column or use the EZP subroutines on 2D slices of 3D grids.





*Code availability.* The EZ Parallel module, the serial and parallel stochastic heat equation solvers, and the serial and parallel two-level QG solvers are available at the GitHub repository

https://github.com/jasonlturner/EZ_PARALLEL_project

## Appendix A: Finite difference code in parallel

In this Appendix, we include excerpts of the parallelized finite difference code, highlighting the changes from the serial version.
Algorithm 4 includes code of the main driver subroutine of the solver. Algorithm 5 includes code for the grid initialization subroutine of the solver. Algorithm 6 includes code for the time-stepping subroutine of the solver.

---

**Algorithm 4** Parallel finite difference solver driver subroutine

---

USE INITIALIZE {The module for reading input files, allocating memory to the grid, setting initial conditions, etc.}

USE TIME_STEPPER {The module for stepping the simulation forward in time}

**USE MPI**

**USE EZ_PARALLEL_STRUCTS** {The EZP module for the parallel scheme, referred to as a scheme within EZP}

**USE EZ_PARALLEL** {The EZP module containing all subroutines of EZ Parallel}

IMPLICIT NONE

**INTEGER :: ierror** {Additional `INTEGER` argument required for all MPI Fortran subroutine calls}

**CALL MPI_INIT(ierror)**

CALL INITIALIZE_PARAMETERS {Reads the `NAMELIST` for various simulation parameters}

CALL **INITIALIZE_GRID** {Allocates memory for the grid, and fills it with the initial condition. Parts of the subroutine need to be changed for parallleization}

CALL **TIME_STEP** {Steps the simulation forward in time and writes output files at the desired frequency. Parts of the subroutine need to be changed for parallelization}

**CALL MPI_FINALIZE(ierror)**

---

**Algorithm 5** Parallel finite difference solver: `INITIALIZE_GRID`

---

**USE MPI**

**USE EZ_PARALLEL_STRUCTS**

**USE EZ_PARALLEL**

IMPLICIT NONE

**CALL CREATE_SCHEME**($N_x$, $N_y$, $\Delta y$, $y_{ref}$, **MPI_COMM_WORLD, MPI_DOUBLE_PRECISION, 1,** $sch$)

{See Message Passing Interface Forum (2015) for information on MPI_COMM_WORLD and MPI_DOUBLE_PRECISION. The "1" argument is the width of sub-grid boundary needed for the time-stepping method, while $sch$ is the scheme for the decomposition}

CALL FILL_INITIAL_CONDITION {Fills in the initial condition for the grid}

---





---

**Algorithm 6** Parallel finite difference solver: `TIME_STEP`

---

USE INITIALIZE

**USE MPI**

**USE EZ_PARALLEL_STRUCTS**

**USE EZ_PARALLEL**

IMPLICIT NONE

CHARACTER(LEN=**14**) :: filename {The length of the filename is changed to include the processor ID, so that each process has a unique output file (see Subsection 2.4)}

**WRITE(filename,'(A,I0.3,A)')** '**output_**', $sch\%procID$, '**.out**' {Set the output file name, now including the processor ID, stored in $sch$ under $procID$}

**for** $i = 1$ to $numTimesteps$ **do**

    CALL TIMESTEP

    CALL UPDATE_GRID {Updates the grid interior}

    **CALL SHARE_SUBGRID_BDRY(grid,** $sch$**)** {Updates the sub-grid boundary of the local sub-grid}

    CALL WRITE_OUTPUT(grid, filename, $i$) {The `WRITE_OUTPUT` subroutine modifies the output file name based on the time-step number}

**end for**

---

*Author contributions.* SNS conceived the idea of the EZ Parallel module and contributed to code design. JLT designed and wrote the EZ Parallel module and example codes. JLT and SNS wrote the paper.

*Competing interests.* The authors declare that they have no competing interests.

*Acknowledgements.* We would like to thank Di Qi for sharing a Matlab implementation of a serial two-level QG solver, and Kevin Welsh for beginning the development of a C++ version of EZ Parallel. We also thank Paul N. Swarztrauber and Hugh C. Pumphrey for the development and availability of FFTPACK and DFFTPACK. This research was performed using the compute resources and assistance of the University of Wisconsin-Madison Center For High Throughput Computing (CHTC) in the Department of Computer Sciences. The CHTC is supported by the University of Wisconsin-Madison, the Advanced Computing Initiative, the Wisconsin Alumni Research Foundation, the Wisconsin

Institutes for Discovery, and the National Science Foundation, and is an active member of the Open Science Grid, which is supported by the National Science Foundation and the U.S. Department of Energy's (DOE) Office of Science. The research of SNS is partially supported by the US Office of Naval Research (ONR) grant N00014-19-1-2421. Funding for JLT comes from the DOE Computational Science Graduate Fellowship, under grant number DE-SC0020347.





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
