# Peer review of "Parallelizing a serial code: open–source module, EZ Parallel 1.0, and geophysics examples"

_Geoscientific Model Development, 2020_

## Short Comment (SC1) · 14 Nov 2020

Dear authors,

in my role as Executive editor of GMD, I would like to bring to your attention our Editorial version 1.2:

https://www.geosci-model-dev.net/12/2215/2019/

This highlights some requirements of papers published in GMD, which is also available on the GMD website in the 'Manuscript Types' section: http://www.geoscientific-model-development.net/submission/manuscript_types.html

[Figure]

In particular, please note that for your paper, the following requirement has not been met in the Discussions paper:

- Code must be published on a persistent public archive with a unique identifier for the exact model version described in the paper or uploaded to the supplement, unless this is impossible for reasons beyond the control of authors. All papers must include a section, at the end of the paper, entitled "Code availability". Here, either instructions for obtaining the code, or the reasons why the code is not available should be clearly stated. It is preferred for the code to be uploaded as a supplement or to be made available at a data repository with an associated DOI (digital object identifier) for the exact model version described in the paper. Alternatively, for established models, there may be an existing means of accessing the code through a particular system. In this case, there must exist a means of permanently accessing the precise model version described in the paper. In some cases, authors may prefer to put models on their own website, or to act as a point of contact for obtaining the code. Given the impermanence of websites and email addresses, this is not encouraged, and authors should consider improving the availability with a more permanent arrangement. Making code available through personal websites or via email contact to the authors is not sufficient. After the paper is accepted the model archive should be updated to include a link to the GMD paper.

As GMD requests persistent access to the exact version of the source code used for the model version presented in the paper, and github is not a persistent archive, please provide for the presented source code a persistent identifier. As explained in https://www.geoscientific-model-development.net/about/manuscript_types.html the preferred reference to this release is through the use of a DOI which then can be cited in the paper. For projects in GitHub a DOI for a released code version can easily be created using Zenodo, see https://guides.github.com/activities/citable-code/ for details.

Finally note, that according to our new Editorial (v1.2) all data and analysis / plotting scripts should be made available.

Yours, Astrid Kerkweg

---

## Author Comment (AC1) · 23 Nov 2020

Dear Editor,

Thank you for taking the time to review our manuscript and pointing out this issue. We have made the following changes to our manuscript:

1) including references to the DOI for the project source code, created through Zenodo.
2) including, as supplements, the scripts and data used to create the plots included in the paper.

These changes will be included in the submission of the revised manuscript. The

authors welcome further constructive comments.

Sincerely,

Jason Turner (jlturner5@wisc.edu)

Samuel Stechmann

---

## Referee Comment (RC1) · Anonymous Referee #1 · 2 Dec 2020

This paper addresses the issue of parallelizing a serial code used in geophysical applications. An open-source module named EZ Parallel is produced for this purpose to automatically with minimum invasion and knowledge of the user parallelize a typical coding setup of geophysical code. The authors demonstrate a very good speed-up of two example codes relative to the amount of user work invested in parallelization. The authors used MPI to achieve this goal.

Mayor point:

In algorithmic and modelling terms, the idea for this paper is not a new thing, but this paper with the supplemented open-source module does bring a merit for giving us a

somewhat simpler way of speeding up a user code on distributed systems, at least for very simple grids. I do like the fact how more advanced usage of MPI becomes hidden in the user code. However, I am concerned about the fact that the overhead of implementing MPI on the user side for simple grids is not that big compared to using EZ Parallel in the code. I would like to be explained more why using EZ Parallel is much easier than MPI for simple grids which you demonstrated. I think this paper would benefit more in explaining the tedious overhead of using MPI over EZ Parallel, in terms of additional lines of code and specific knowledge of MPI needed. Also assume we have a user who just coded two-level quasi-geostropic equation and has a pretty good idea how to use MPI to parallelise it. Why EZ Parallel is much better for him?

General questions:

1. Have you investigated using asynchronous MPI? Is there any problem adding support for OpenMP through EZ Parallel?

2. After parallelising the code and its output, one is faced with multiple output_procID which need to merged and visualized. Can EZ Parallel provide a tool to that, or is merging left to the user?

Minor remarks:

line 78: "with modification in red" but I do not see red text in Algorithm 2

line 195: andf -> and

line 211: Did you test scaling with I/O?

line 257: psuedo -> pseudo

Algorithm 4 after INITIALIZE_GRID: paralllelization -> parallelization

Fig 6 & Fig 7 take a large portion but are not directly relevant, they could be grouped together in Fig 6a 6b.

---

## Referee Comment (RC2) · Anonymous Referee #2 · 23 Dec 2020

This paper presents the authors' efforts on developing a tool that can help the process of parallelizing a serial code, at the level of MPI-based parallelization.

This is a traditionally interesting topic, even more interesting at the current stage of migrating from homogeneous multi-core clusters to heterogeneous clusters with GPUs or other many-core accelerators.

This is overall a quite interesting work. The authors have also done a good job to provide not only a tool but also demonstrations in a number of different applications or kernels.

My major question is about the comparison with other similar efforts in the field. Parallelization, especially MPI-based parallelization, has been around for many decades. Groups from both computer science and application domains have existing projects that try to derive languages, compiler, tools to support better and easier parallelization. Therefore, an introduction of such a tool should come with a comprehensive overview of existing efforts. Also, for demonstrating the efficiency and performance of the proposed tool, comparisons should be made on both the parallel performance achieved, and the extra coding efforts needed. In the current paper, we only see results of the parallel code in the proposed tool, but not sure how good it is when compared to other similar tools, or languages, such as the Unified Parallel C project from Berkeley.

Another minor complaint is about the lack of support for parallelism at thread level. My understanding is that the backbone of the tool is based on MPI, which is still the best way to go for parallelization across different nodes. But the current hardware trend is to have more and more computing power within a node, and you can easily have a heavy CPU with around 50 cores, or a GPU with thousands of CUDA cores. It would be a more interesting tool if these cases can be covered, or at least discussed. For example, for a node with around 100 cores, how would such a tool perform against an OpenMP approach?

---

## Author Comment (AC2) · 8 Jan 2021

Dear Anonymous Referee,

Thank you for reading our paper and providing these helpful comments. Below are detailed responses to your comments. *Your comments are shown in black italics*, the authors' response is in blue, and the authors' changes in the manuscript are in green.

*This paper addresses the issue of parallelizing a serial code used in geophysical applications. An open-source module named EZ Parallel is produced for this purpose to automatically with minimum invasion and knowledge of the user parallelize a typical*

*coding setup of geophysical code. The authors demonstrate a very good speed-up of two example codes relative to the amount of user work invested in parallelization. The authors used MPI to achieve this goal.*

*Mayor point:*

*In algorithmic and modelling terms, the idea for this paper is not a new thing, but this paper with the supplemented open-source module does bring a merit for giving us a somewhat simpler way of speeding up a user code on distributed systems, at least for very simple grids. I do like the fact how more advanced usage of MPI becomes hidden in the user code. However, I am concerned about the fact that the overhead of implementing MPI on the user side for simple grids is not that big compared to using EZ Parallel in the code. I would like to be explained more why using EZ Parallel is much easier than MPI for simple grids which you demonstrated. I think this paper would benefit more in explaining the tedious overhead of using MPI over EZ Parallel, in terms of additional lines of code and specific knowledge of MPI needed. Also assume we have a user who just coded two-level quasi-geostropic equation and has a pretty good idea how to use MPI to parallelise it. Why EZ Parallel is much better for him?*

A new paragraph has been added to explain how using EZP can be easier than directly writing out the MPI commands. Other new description, in addition to this paragraph, is also aimed at explaining why EZP could be useful for a user who already has experience with MPI.

One benefit to such a user would be the included parallel fast Fourier transforms that are included in EZ Parallel, as they utilize an algorithm introduced by Mortensen et. al. in their 2019 paper "Fast Parallel Multidimensional FFT Using Advanced MPI" (https://doi.org/10.1016/j.jpdc.2019.02.006). The EZ Parallel fast Fourier transform can handle 1-D and 2-D transforms of both double precision and double precision complex data types, using approximately 350 lines of code (including initializing the transforms and error handling). The EZ Parallel fast Fourier transform requires only two subroutine calls by the user, a call to the transform initialization and a call to the execution of the transformation. Additionally, the subroutines of EZ Parallel have been thoroughly tested and debugged, and will notify the user of many different types of errors. MPI itself has a relatively poor repertoire of easily-accessible and easily-usable debugging tools, so they could use this library with confidence that it would not readily introduce errors into their code.

(LINES 44 - 57)

The goal of EZP is to break the parallelization process into a few high-level tasks. For the scenarios of interest described above, where the setup involves PDEs on a 2D or 3D domain, the high-level tasks include, for instance, a domain decomposition to break the domain into several partitions (see Section 2). For each of the high-level tasks, the goal of EZP is to provide a corresponding subroutine. Then the subroutine can be called to accomplish the task, and the user does not need to worry about the parallel programming details. For instance, a simple command `CALL DOMAIN_DECOMPOSITION` could be inserted into a code, and the domain decomposition will be accomplished without the user needing to write their own parallel programming algorithms. The EZ Parallel subroutine that accomplishes the domain decomposition contains approximately 150 lines of code, including defining MPI derived datatypes to be used in some inter-core communications and error handling. As an analogous situation that is already in widespread use, consider the calculation of Fourier transforms. Most users do not write their own Fourier transform algorithms, in parallel or in serial. Instead, a user would typically take advantage of the many established Fourier transform codes that have been written, which allow the user to type a simple command such as `y = FFT(x)` and obtain the Fourier transform. No knowledge is needed of the underlying details of the Fourier transform code or algorithm. The goal of EZP is to provide a similar type of high-level functionality for some parallel programming tasks, in order to ease the process of parallelizing a serial code.

(LINES 98 - 100)

Also, we hope that even experienced users of MPI will find the EZP module useful, since EZP could speed up and simplify the code-writing process (by providing high-level functionality via the grouping together of MPI commands).

*General questions:*

*1. Have you investigated using asynchronous MPI? Is there any problem adding support for OpenMP through EZ Parallel?*

We have indeed considered using asynchronous MPI, and it would undoubtedly allow the user to further optimize the performance of their code. However, we would like to strike a balance between providing the user a variety of tools and simplifying the process of parallelizing their serial code. Since we would like EZ Parallel to be easy for users who have relatively little experience with parallel computing, for simplicity's sake we have utilized only blocking sends/receives. In future versions of the EZ Parallel, we hope to find a simplistic way of implementing subroutines with asynchronous MPI.

We also did not utilize OpenMP for a similar reason, and hope to provide functionality with OpenMP in future versions.

(LINES 313 - 314)

It would also be interesting to explore asynchronous MPI, or to pursue similar software involving CUDA or OpenMP, or other languages such as Julia or Python.

*2. After parallelising the code and its output, one is faced with multiple output_procID which need to merged and visualized. Can EZ Parallel provide a tool to that, or is merging left to the user?*

The EZ Parallel examples come with example Matlab scripts that we used for merging output files and visualizations.

(LINES 108 - 109)

In the examples of parallelized codes we have provided example Matlab scripts that
we used for merging output files (see Subsection 2.4) and visualizations.

*Minor remarks:*

*line 78: "with modification in red" but I do not see red text in Algorithm 2*

Thank you – this has been corrected.

The text "with modifications in red" has been changed to "with modifications in bold-face".

*line 195: andf − > and*

Thank you – this typo has been corrected.

*line 211: Did you test scaling with I/O?*

I/O was disabled during the scaling tests, following somewhat standard practice.

(LINES 249 - 250)

File output was disabled for all tests following common practice (e.g., Arabas et. al., 2015), and we analyzed the average execution time per time-step for consistency between runs.

*line 257: psuedo − > pseudo*

Thank you – this typo has been corrected.

*Algorithm 4 after INITIALIZE_GRID: paralllelization − > parallelization*

Thank you – this typo has been corrected.

*Fig 6 & Fig 7 take a large portion but are not directly relevant, they could be grouped together in Fig 6a 6b.*

It is true that these two figures take up a large amount of space. If we were to shrink the size of the figures, though, then the serial-simulation domain (shown in the inset)

would become so small that it would be difficult to discern. We would like to maintain our original intent for these figures, which was to highlight the much bigger domain size that can be utilized in parallel simulations versus serial simulations, and we do feel that a figure highlights this nicely. To keep this intent and also save space, we have taken the following approach.

Given that the intent can be accomplished with only Figure 6 by itself, we have removed Figure 7.

We hope that these responses are satisfactory and would like to again thank you for your efforts and constructive feedback. The authors welcome any further constructive comments.

Sincerely,

Jason Torchinsky [Formerly "Turner"] (jlturner5@wisc.edu)

Samuel Stechmann

---

## Author Comment (AC3) · 8 Jan 2021

Dear Anonymous Referee,

Thank you for reading our paper and providing these helpful comments. Below are detailed responses to your comments. *Your comments are shown in black italics*, the authors' response is in blue, and the authors' changes in the manuscript are in green.

*This paper presents the authors' efforts on developing a tool that can help the process of parallelizing a serial code, at the level of MPI-based parallelization. This is a traditionally interesting topic, even more interesting at the current stage of migrating*

*from homogeneous multi-core clusters to heterogeneous clusters with GPUs or other many-core accelerators.*

*This is overall a quite interesting work. The authors have also done a good job to provide not only a tool but also demonstrations in a number of different applications or kernels.*

*My major question is about the comparison with other similar efforts in the field. Parallelization, especially MPI-based parallelization, has been around for many decades. Groups from both computer science and application domains have existing projects that try to derive languages, compiler, tools to support better and easier parallelization. Therefore, an introduction of such a tool should come with a comprehensive overview of existing efforts. Also, for demonstrating the efficiency and performance of the proposed tool, comparisons should be made on both the parallel performance achieved, and the extra coding efforts needed. In the current paper, we only see results of the parallel code in the proposed tool, but not sure how good it is when compared to other similar tools, or languages, such as the Unified Parallel C project from Berkeley.*

Now included in the revised manuscript are four lengthy paragraphs, for a total of about 1.5 pages of text, to provide an overview of and comparisons with existing efforts. The first of the four paragraphs is new and is intended to describe the goals of EZP, for contrast with other existing efforts. The second of the four paragraphs was mostly present in the original manuscript, but it has been expanded and modified in the revised manuscript, and it describes efforts for automatic parallelization. The third of the four paragraphs is another new paragraph, and it focuses on interactive parallelization tools. Lastly, the fourth of the four paragraphs is a new paragraph that focuses on alternatives to MPI such as Unified Parallel C and Coarray Fortran. In these paragraphs, we have tried to more clearly describe the goal of EZP to create a parallel code that looks as similar as possible to the original serial code, by grouping together the MPI commands into a few subroutines calls.

The goal of EZP is to break the parallelization process into a few high-level tasks. For the scenarios of interest described above, where the setup involves PDEs on a 2D or 3D domain, the high-level tasks include, for instance, a domain decomposition to break the domain into several partitions (see Section 2). For each of the high-level tasks, the goal of EZP is to provide a corresponding subroutine. Then the subroutine can be called to accomplish the task, and the user does not need to worry about the parallel programming details. For instance, a simple command `CALL DOMAIN_DECOMPOSITION` could be inserted into a code, and the domain decomposition will be accomplished without the user needing to write their own parallel programming algorithms. The EZ Parallel subroutine that accomplishes the domain decomposition contains approximately 150 lines of code, including defining MPI derived datatypes to be used in some inter-core communications and error handling. As an analogous situation that is already in widespread use, consider the calculation of Fourier transforms. Most users do not write their own Fourier transform algorithms, in parallel or in serial. Instead, a user would typically take advantage of the many established Fourier transform codes that have been written, which allow the user to type a simple command such as `y = FFT(x)` and obtain the Fourier transform. No knowledge is needed of the underlying details of the Fourier transform code or algorithm. The goal of EZP is to provide a similar type of high-level functionality for some parallel programming tasks, in order to ease the process of parallelizing a serial code.

A related but different type of tool is automatic parallelization, which is intended to take a serial code and, without needing the user to change the code at all, automatically implement parallelization in its execution (Griebl, 2004; Bondhugula et al., 2008; Benabderrahmane et al., 2010; Kraft et al., 2018). Such tools are typically designed to treat a variety of different codes with great generality. For instance, the code could be for playing the game of chess, or some other artificial intelligence application. In their simplest forms, auto-parallelizers detect "for loops" in the code and, if possible, they

break up the tasks using multi-threading or parallelization. Some freely available tools are Cetus (Dave et al., 2010), ComPar (Mosseri et al., 2020), Par4All (Amini et al., 2012), and Polaris (Blume et al., 1994). Some commercial tools that use these and other strategies include the Matlab Parallel Computing Toolbox™ and Matlab Parallel Server™ or Matlab Distributed Computing Server™. A disadvantage of Matlab is that it is often slower than other programming languages (Fortran, C++, etc.) for PDEs with time-stepping. For those other languages, the Intel® Fortran and C++ Compilers also have automatic parallelization capabilities, and so does the freely available GNU Compiler Collection (GCC). The commercial tools are not freely available and therefore may not be an option for many users. Auto parallelization is typically limited to shared-memory computer architectures, which limits the amount of parallelization that can be achieved. Also, for some codes, auto parallelization may not achieve parallelism or may not provide a great amount of speedup. Despite these limitations, it may be a good choice for some users. However, some users may also have concerns about using a "black box" to automatically parallelize a code. Some users prefer to have some control over and some knowledge of the parallelization process. For such users, EZP can help with parallelization with a minimal amount of user effort, while also engaging the user in the parallelization process, and allowing the user to further optimize the parallelization if desired.

Besides fully automatic tools, other tools have been created to be more interactive but still somewhat automatic. Examples include ParaScope (Balasundaram et al., 1989), the Parallelizing Assistant Tool (PAT) (Appelbe and Smith, 1990), FORGE (Levesque and Wagenbreth, 2011), and the Interactive Parallelization Tool (IPT) (Arora et al., 2014). The interactive tools can provide valuable feedback to a user and can help the user learn more about parallelization, in comparison to automatic tools which are typically not designed for such purposes. Like automatic tools, the interactive tools are commonly designed to be very general. In the present paper, it is not the general case of parallelization that is considered, but the specific case of parallelization of PDE solvers using domain decomposition. In this specific case, following the methods de-
scribed below, a desirable feature can be achieved: the parallel code looks essentially the same as the original serial code. One of the main differences from the original serial code is the addition of some new subroutine calls, since much of the MPI functionality can be grouped together into a few subroutines. In this way, one goal of EZP is to preserve the readability of the code as much as possible.

As another route to simplifying parallel coding, some alternatives to MPI and OpenMP have been developed as parallel languages that extend serial programming languages such as C and Fortran. Examples that use the global address space (GAS) model include Unified Parallel C (UPC) (Chen et al., 2003) and Coarray Fortran (Numrich and Reid, 1998). These examples aim to offer a programming environment that is more user-friendly compared with the message passing of MPI. It would be interesting to try to include UPC and/or Coarray Fortran capabilities into EZP instead of MPI, and to try to achieve the same goals as MPI-based EZP—namely, a parallel code that looks nearly identical to (or as similar as possible to) the original serial code, aside from a few subroutine calls to handle the parallelization. For this initial work, MPI was chosen in part because it is in more widespread use.

*Another minor complaint is about the lack of support for parallelism at thread level. My understanding is that the backbone of the tool is based on MPI, which is still the best way to go for parallelization across different nodes. But the current hardware trend is to have more and more computing power within a node, and you can easily have a heavy CPU with around 50 cores, or a GPU with thousands of CUDA cores. It would be a more interesting tool if these cases can be covered, or at least discussed. For example, for a node with around 100 cores, how would such a tool perform against an OpenMP approach?*

Thank you for these suggestions. It would indeed be interesting to include CUDA capabilities into EZP, and this is now mentioned in the revised manuscript. Also, new results have been added to show the scaling performance on a single node with a large number of processors. We have access to a single node with 64 cores.

New Figure 5 and new description single-node scaling performance (LINES 239 - 259)

We performed two series of tests to investigate the scaling performance of the implementation of the EZP library to parallelize a serial two-level QG equation code: across multiple nodes with several cores per node and on a single node with several cores. We conducted the multi-node tests on the high performance computing (HPC) cluster that is serviced by the University of Wisconsin-Madison's Center for High Throughput Computing (CHTC) and the single-node tests on the Cori System that is serviced by the National Energy Research Scientific Computing Center (NERSC). For the multi-node tests, we utilized a dedicated partition equipped with 35 nodes, each with a single 10-core (20 thread) Intel Xeon E5-2670 v2 CPU at 2.50 GHz. For the single-node tests, we utilized the KNL partition, whose nodes are a single-socket Intel Xeon Phi Processor 7250 processor with 68 cores at 1.4 GHz.

Simulations were run with a random initial condition on a grid with dimensions $N_x = N_y = 2048$ and $10^2$ time-steps. The physical parameters in the simulation correspond to the atmosphere in mid-latitudes, specifically: $\beta = 2.5$, $k_d = 4$, $U = 0.2$, $\kappa = 0.05$, and $\nu = 2.98023 \times 10^{-22}$ (Qi and Majda, 2016). File output was disabled for all tests following the practice of Arabas et al. (2015), and we analyzed the average execution time per time-step for consistency between runs. The multi-node tests used a range of two to 128 cores, utilizing eight cores per node for runs of eight or more cores. We chose not to utilize all 20 available cores per node due to the known lack of optimization of MPI_ALLTOALL on shared-memory systems (Kumar et al., 2008). Due to the memory-intensive nature of the six-stage AARK method used, there was not enough memory available for a single-core run with $N_x = N_y = 2048$. The single-node tests used a range of one to 64 cores on a single node. The results of these tests are included in Figs. 4 and 5.

We observed a strong scaling efficiency of $73.0\% \pm 7.8\%$ for the multi-node tests and a strong scaling efficiency of $94.8\% \pm 11.1\%$ for the single-node tests, which implies that doubling the number of cores across several nodes leads to an approximate $40\%$

reduction in execution time and double the number of cores across a single node leads to an approximate $48\%$ reduction in the execution time.

(LINES 313 - 314)

It would also be interesting to explore asynchronous MPI, or to pursue similar software involving CUDA or OpenMP, or other languages such as Julia or Python.

We hope that these responses are satisfactory and would like to again thank you for your efforts and constructive feedback. The authors welcome any further constructive comments.

Sincerely,

Jason Torchinsky [Formerly "Turner"] (jlturner5@wisc.edu)

Samuel Stechmann